# Immuno-VLM: Immunizing Large Vision-Language Models via Generative Semantic Antibodies for Open-World Trustworthiness

Xiang Fang [1]    Wanlong Fang [2]    Wei Ji [3]

## Abstract

Large Vision-Language Models have achieved unprecedented success in zero-shot recognition by aligning visual features with broad semantic concepts. However, this semantic abstraction creates a critical vulnerability in open-world deployment: the "Hubris of Semantics", where models force-fit unknown anomalies into known categories with high confidence due to the lack of explicit negative knowledge. To address this *Open-World Trustworthiness Paradox*, we propose **Immuno-VLM**, a bio-inspired framework that adapts the biological principle of **Immunological Negative Selection** to high-dimensional latent spaces. Departing from traditional Open-Set Recognition methods that rely on passive density estimation or inefficient pixel-space outlier generation, Immuno-VLM leverages the generative reasoning of Large Language Models to actively hallucinate "Semantic Antibodies", textual descriptions of near-distribution outliers (e.g., look-alikes, contextual anomalies) that effectively bound the decision space of known classes. Extensive experiments on ImageNet-1K and four challenging OOD benchmarks reveal that Immuno-VLM establishes a new state-of-the-art.

## 1. Introduction

The paradigm of computer vision has been irrevocably altered by the emergence of Large Vision-Language Models (LVLMs) (Cai et al., 2025; Fang et al., 2022; 2026d; Lei et al., 2025; Fang et al., 2023b; Wang et al., 2025a), such as CLIP (Radford et al., 2021), ALIGN (Jia et al., 2021), and LLaVA (Liu et al., 2023d). By scaling contrastive pre-

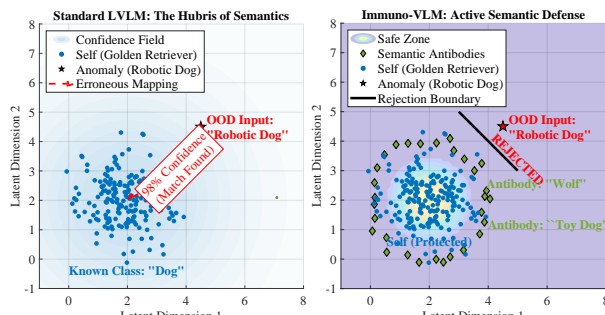

*Figure 1.* The Core Concept. While standard LVLMs suffer from the "Hubris of Semantics", force-fitting anomalies into known classes, Immuno-VLM uses hallucinated semantic antibodies to actively define the boundary of the unknown.

training to billions of noisy image-text pairs, these foundation models have moved beyond the closed-set limitations of traditional supervised learning, acquiring a remarkable ability to generalize to novel visual concepts via natural language prompts (Bommasani et al., 2021; Fang, 2026; Fang et al., 2026e; Wang et al., 2026b; Liu et al., 2023c; 2026; Fang et al., 2026f; Wang et al., 2025c; Fang et al., 2026g; 2025g; 2024c; Liu et al., 2024c; Fang et al., 2025f;d; 2024b). This "zero-shot" capability suggests a future where AI systems can operate robustly in the unconstrained open world (Fang et al., 2025b;e; Yan et al., 2026; Fang et al., 2025a; Wang et al., 2026c; Fang & Fang, 2026b; Wang et al., 2026a; Fang & Hu, 2020; Wang et al., 2026d; Fang & Fang, 2026a; Wang et al., 2025f). However, as these models transition from academic benchmarks to safety-critical deployments, ranging from autonomous driving to medical diagnostics, a fundamental epistemic vulnerability has emerged: the *Open-World Trustworthiness Paradox*.

While LVLMs demonstrate impressive robustness to covariate shifts (e.g., recognizing a sketch of a dog as easily as a photo) (Radford et al., 2021; Liu et al., 2023b; Fang et al., 2024d; Liu et al., 2024a; Fang et al., 2023a; Xiong et al., 2024), they exhibit a perilous fragility when distinguishing "known" concepts from "unknown" anomalies. We term this the "Hubris of Semantics". Because models like CLIP are trained to maximize the cosine similarity between visual embeddings and the nearest text concepts in a dense semantic manifold (Wang & Isola, 2020; Fang et al., 2021b; Wang et al., 2025e; Zhang et al., 2025a; Fang et al., 2026a;

[1]School of Software Engineering, Huazhong University of Science and Technology [2]Nanyang Technological University, Singapore [3]Nanjing University. Correspondence to: Wanlong Fang <wanlongfang@gmail.com>.

*Proceedings of the 43rd International Conference on Machine Learning*, Seoul, South Korea. PMLR 306, 2026. Copyright 2026 by the author(s).

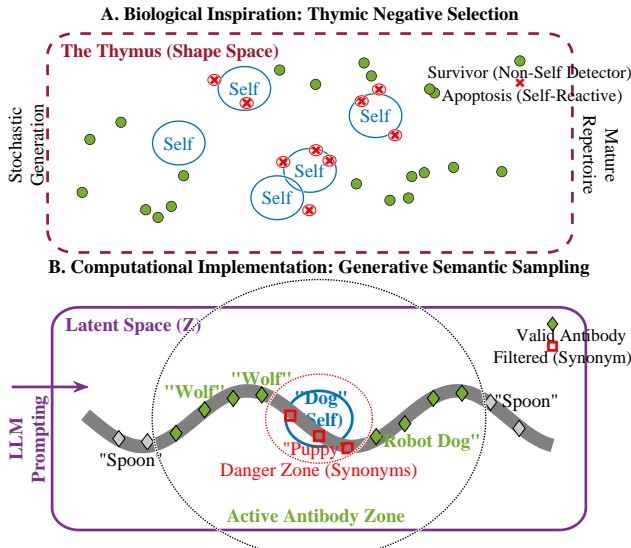

*Figure 2.* The Isomorphism between Biological Immunity and Immuno-VLM. We map T-cell generation to Antibody Hallucination and Thymic Selection to our Active Density Filter.

Tang et al., 2024; 2025; Fang et al., 2021a; Cai et al., 2026; Fang et al., 2020), they develop a retrieval bias that forces virtually any visual input into a known category. When confronted with an out-of-distribution (OOD) anomaly, such as a "robotic dog" or an adversarial texture (Hendrycks et al., 2021), the model does not confess ignorance. Instead, it hallucinates a high-confidence match based on superficial feature overlaps, classifying the robot as a "Golden Retriever" with near-certainty. This behavior represents a regression from the principles of Open Set Recognition (OSR) established by (Scheirer et al., 2012), where the ability to reject the "unknown" is paramount.

Traditional defenses against Open Space Risk have largely relied on discriminative thresholding. Methods such as Maximum Softmax Probability (MSP) (Hendrycks & Gimpel, 2017), Energy Scores (Liu et al., 2020), and Activation Shaping (Djurisic et al., 2023) attempt to identify anomalies by analyzing the magnitude or distribution of activation vectors. While effective in low-dimensional spaces, these methods struggle in the high-dimensional latent spaces of Foundation Models due to the concentration of measure phenomenon (Wang & Isola, 2020), where the "open space" becomes vast and sparsely populated. More recent approaches like Maximum Concept Matching (MCM) (Ming et al., 2022) attempt to leverage the text modality, but they remain *reactive*, waiting for an error to manifest as a statistical deviation, rather than *proactive*. Furthermore, generative approaches that attempt to synthesize pixel-space outliers using GANs (Ge et al., 2017; Neal et al., 2018) face a combinatorial explosion when scaled to the diversity of ImageNet, rendering them computationally intractable.

To resolve this paradox, we look to the biological immune

system, the only known learning system that successfully solves the open-world recognition problem at a planetary scale (Dasgupta, 1999). The vertebrate immune system protects the organism from a virtually infinite space of pathogens (viruses, bacteria, tumors), including those that have not yet evolved, without requiring prior exposure to them. It achieves this feat not by memorizing the external world, which is impossible, but by rigorously defining the "Self". Through a process called *Negative Selection* in the thymus (Forrest et al., 1994), the body generates millions of random T-cell receptors. These candidates are tested against self-antigens; those that bind to "Self" are destroyed (clonal deletion), while the non-reactive survivors are released into the bloodstream. These survivors essentially define a "Repertoire Hole", a negative image of the Self (Jerne, 1974). Consequently, if a T-cell binds to *anything* in the periphery, the system can mathematically guarantee that the target is an anomaly ("Non-Self"), even if that specific pathogen has never been encountered before.

In this work, we propose **Immuno-VLM**, a framework that translates this biological principle into a rigorous computational algorithm for large-scale vision models. We address the "Curse of Dimensionality" that plagued early Artificial Immune Systems (Forrest et al., 1994) by shifting the domain of negative selection from the pixel space to the *Semantic Manifold*. Instead of generating random detectors, we leverage the reasoning capabilities of Large Language Models (LLMs) like GPT-4 (Vaswani et al., 2017) to act as a "Computational Thymus". The LLM actively hallucinates "Semantic Antibodies", textual descriptions of "Near-OOD" concepts (e.g., "wolf", "coyote", "robot dog" for the class "dog") and "Contextual Anomalies" (e.g., "dog melting", "dog made of clouds"). These descriptions are projected into the shared embedding space to form a dense, semantically informed boundary around the known classes.

Our contributions are threefold. **Methodologically**, we introduced the concept of **Generative Semantic Antibodies**, demonstrating that Large Language Models can serve as a "Computational Thymus", hallucinating a vast, diverse repertoire of near-distribution outliers that pixel-space generators cannot feasibly produce. **Theoretically**, we derived the *Antibody Covering Bound* (Theorem 3.3) and the *Manifold Sample Complexity* analysis, providing the first rigorous mathematical proof that targeting the *Semantic Manifold* breaks the Curse of Dimensionality that has historically plagued Artificial Immune Systems. We showed that the safety of a vision system is strictly bounded by the imagination (Wasserstein alignment) of its language generator. **Empirically**, Immuno-VLM achieved state-of-the-art performance on the demanding multiple challenging benchmarks, improving the detection of adversarial semantic shifts by over 16% compared to standard zero-shot baselines, while preserving the integrity of in-distribution recognition.

## 2. Related Work

As a challenging machine learning task (Liu et al., 2023a; Wang et al., 2025d; Fang et al., 2026b; Kuai et al., 2026; Wang et al., 2025b; Fang et al., 2025c; Zhang et al., 2025b; Fang et al., 2023c; Liu et al., 2024b; Yang et al., 2025), the out-of-distribution (OOD) detection task aims to detect test samples from distributions that do not overlap with the training distribution. Previous OOD detection methods (Liang et al., 2018; Liu et al., 2020; Sun et al., 2021; Lee et al., 2018b; Mohseni et al., 2020; Vyas et al., 2018; Yu & Aizawa, 2019; Zaeemzadeh et al., 2021; Hsu et al., 2020; Ming et al., 2023; Jiang et al., 2024) can be divided into four types: classification-based methods (Hendrycks & Gimpel, 2016; Liang et al., 2018; Lee et al., 2018c;a), density-based methods (Kirichenko et al., 2020; Serrà et al., 2019), distance-based methods (Techapanurak et al., 2020; Lee et al., 2018b) and reconstruction-based methods (Zhou, 2022; Yang et al., 2022).

## 3. The Proposed Immuno-VLM Framework

### 3.1. Problem Formulation

We consider the problem of open-world visual recognition within a joint vision-language embedding space. Let $\mathcal{X} \subseteq \mathbb{R}^{H \times W \times 3}$ denote the input image space and $\mathcal{T}$ denote the discrete space of natural language descriptions.

We assume the existence of a pre-trained Vision-Language Model (VLM) consisting of dual encoders: a visual encoder $\phi_v : \mathcal{X} \to \mathcal{Z}$ and a text encoder $\phi_t : \mathcal{T} \to \mathcal{Z}$, mapping both modalities into a shared hyperspherical latent space $\mathcal{Z} \subset \mathbb{R}^d$, where typically $||z||_2 = 1$ for all $z \in \mathcal{Z}$.

**The Open-World Risk Landscape** In a standard Closed-Set setting, the training data $\mathcal{D}_{train} = \{(x_i, y_i)\}_{i=1}^N$ is drawn from a joint distribution $P_{\mathcal{X}\mathcal{Y}}$ defined over a known label set $\mathcal{Y}_{in} = \{1, \dots, K\}$. The goal is to minimize the empirical risk $\hat{\mathcal{R}}_{in}(f)$.

In the Open-World setting, we encounter a test distribution $P_{test}$ which is a mixture of the known distribution $P_{in}$ and an unknown distribution $P_{out}$ defined over a label set $\mathcal{Y}_{out}$, where $\mathcal{Y}_{in} \cap \mathcal{Y}_{out} = \emptyset$. The crucial challenge is that the support of $P_{out}$ in the pixel space $\mathcal{X}$ is unbounded. We define the *Open Space Risk* $\mathcal{R}_{\mathcal{O}}(f)$ following the formulation by Scheirer et al., but adapted for the metric space $\mathcal{Z}$:

**Definition 3.1** (**Metric Open Space Risk**). Given a recognition function $f : \mathcal{Z} \to \mathcal{Y}_{in} \cup \{\perp\}$ (where $\perp$ denotes rejection), the open space risk is defined as the probability of misclassifying an open-world sample as a known class, weighted by its distance from the known distribution:

$$\mathcal{R}_{\mathcal{O}}(f) = \int_{\mathcal{O}} \mathbb{I}(f(z) \in \mathcal{Y}_{in}) \cdot \nu(z) \, dz, \qquad (1)$$

where $\mathcal{O} = \mathcal{Z} \setminus \text{supp}(P_{in})$ is the open space, $\mathbb{I}(\cdot)$ is the indicator function, and $\nu(z)$ is a probability measure of observing a sample at $z$.

The fundamental difficulty in minimizing $\mathcal{R}_{\mathcal{O}}(f)$ is that $\mathcal{O}$ is vast. Standard methods try to bound $\text{supp}(P_{in})$ tightly (e.g., using hyperspheres). However, without explicit samples from $\mathcal{O}$, these boundaries are often arbitrary and loose.

**The Semantic Manifold Assumption** To address the intractability of sampling from the entire open pixel space, we leverage the structure of the VLM. We posit that the open space $\mathcal{O}$ is not structurally random but follows a *semantic manifold*.

**Assumption 3.2** (**Bi-Lipschitz Semantic Alignment**). We assume the text encoder $\phi_t$ and visual encoder $\phi_v$ share an alignment structure such that for any visual concept $x$ and its true semantic description $t$, there exists a constant $L$ such that the embedding distance is bounded:

$$\|\phi_v(x) - \phi_t(t)\|_2 \leq \epsilon_{align}. \qquad (2)$$

Furthermore, the mapping from semantic concepts to visual embeddings is locally Bi-Lipschitz. Let $S_{concept}$ be a semantic space metric. For distinct concepts $t_1, t_2$:

$$c_1 S_{concept}(t_1, t_2) \leq \|\phi_t(t_1) - \phi_t(t_2)\|_2 \leq c_2 S_{concept}(t_1, t_2).$$

It implies that if we can identify textual descriptions (Antibodies) that are semantically distinct from $\mathcal{Y}_{in}$ but semantically close to the boundary of $\mathcal{Y}_{in}$, their projections in $\mathcal{Z}$ will structurally bound the visual open space.

**Semantic Antibodies and Covering Bound** We define a set of **Semantic Antibodies** $\mathcal{A} = \{a_1, \dots, a_M\} \subset \mathcal{T}$ as a set of generated descriptions such that for all $a_j \in \mathcal{A}$ and $y \in \mathcal{Y}_{in}$, $a_j$ is semantically distinct from $y$. Let $H_k \subset \mathcal{Z}$ be the hypothesis space (acceptance region) for known class $k$. The goal of **Immuno-VLM** is to shape $H_k$ such that it minimizes overlaps with the antibody embeddings $\phi_t(\mathcal{A})$.

**Theorem 3.3** (**The Antibody Covering Bound**). *Let $\mathcal{A}_\delta$ be a $\delta$-cover of the semantic boundary of $\mathcal{Y}_{in}$, meaning for any open-world concept $z_{out}$ within distance $\gamma$ of a known class prototype $\mu_k$, there exists an antibody $a \in \mathcal{A}_\delta$ such that $\|z_{out} - \phi_t(a)\| \leq \delta$. If the classifier $f$ enforces a margin $m > \delta$ between known prototypes and all antibodies in $\mathcal{A}_\delta$, then the False Positive Rate (FPR) on open-world samples in the $\gamma$-neighborhood is strictly bounded by the alignment error $\epsilon_{align}$.*

*Proof.* Let $x_{out}$ be an unknown sample with true semantic description $t_{out}$. By Assumption 3.2, $\|\phi_v(x_{out}) - \phi_t(t_{out})\| \leq \epsilon_{align}$. Since $\mathcal{A}_\delta$ is a cover, there exists an antibody $a^* \in \mathcal{A}_\delta$ such that $\|\phi_t(t_{out}) - \phi_t(a^*)\| \leq \delta$. Using the triangle inequality: $\|\phi_v(x_{out}) - \phi_t(a^*)\| \leq \|\phi_v(x_{out}) - \phi_t(t_{out})\| + \|\phi_t(t_{out}) - \phi_t(a^*)\| \leq \epsilon_{align} + \delta$. The classifier $f$ rejects

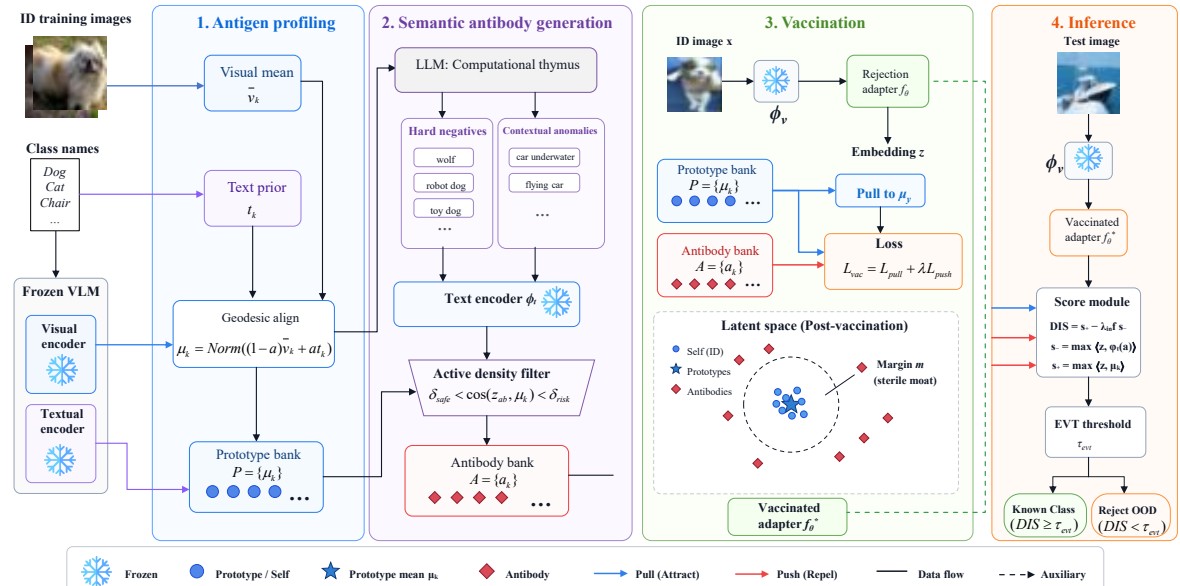

Figure 3. The Immuno-VLM Framework. The pipeline transforms a pre-trained VLM into a trustworthy open-world recognizer by explicitly defining the "Non-Self" space via generative hallucination.

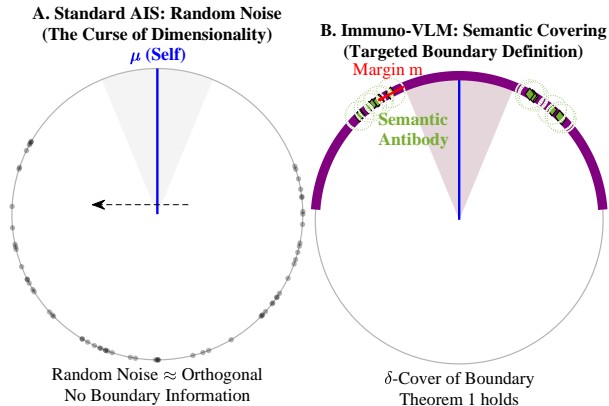

Figure 4. Breaking the Curse of Dimensionality. By targeting the semantic manifold, a finite number of antibodies ($M \approx 100$) can effectively cover the boundary of the Self, whereas random sampling in $\mathbb{R}^{512}$ would require exponentially more points.

$x$ if its similarity to any known prototype $\mu_k$ is lower than a threshold derived from the antibodies. If we enforce the training constraint (Vaccination Loss) such that prototypes are separated from antibodies by margin $m$: $\|\mu_k - \phi_t(a^*)\| \geq m$. For a false positive to occur, $x_{out}$ must be close to $\mu_k$. However, $x_{out}$ is anchored to $a^*$. If $m > \epsilon_{align} + \delta$, the triangle inequality ensures $x_{out}$ cannot be arbitrarily close to $\mu_k$ without violating the margin constraint or the metric structure. Thus, the risk is bounded by the density of the cover $\delta$ and the alignment quality $\epsilon_{align}$. $\square$

This theorem provides the theoretical foundation for our method: optimizing against a dense set of LLM-generated near-OOD descriptions effectively tightens the decision boundary in a way that random noise injection (standard

AIS) cannot.

## 3.2. Method Overview

Our proposed Immuno-VLM framework, illustrated in Figure 3, operationalizes the biological principle of negative selection through a three-phase pipeline designed to transform a pre-trained Vision-Language Model into a trustworthy open-world recognizer.

## 3.3. Phase 1: Antigen Profiling via Bayesian Hyperspherical Estimation

In the biological immune system, the definition of "Self" is established via the presence of self-antigens. In our computational framework, the "Self" for class $k$ is not merely a single point but a distribution on the hypersphere $\mathcal{Z}$. Standard approaches often approximate this using the empirical mean of visual features. However, LVLMs suffer from a *modality gap*, where the visual centroid and the text embedding of the class name are misaligned. To rigorously define the Antigen Profile $\mathbf{p}_k$ for each class, we propose a **Text-Regularized Von Mises-Fisher (vMF) Estimation**.

**Hyperspherical Distribution Modeling** Since the embeddings are $L_2$-normalized, the feature space is the unit hypersphere $\mathbb{S}^{d-1}$. We model the distribution of "Self" features for class $k$ using a Von Mises-Fisher distribution $vMF(\boldsymbol{\mu}_k, \kappa_k)$, with probability density function:

$$f(\mathbf{z}|\boldsymbol{\mu}_k, \kappa_k) = C_d(\kappa_k) \exp(\kappa_k \boldsymbol{\mu}_k^\top \mathbf{z}), \quad (3)$$

where $\boldsymbol{\mu}_k \in \mathbb{S}^{d-1}$ is the mean direction (the Antigen Prototype), $\kappa_k \geq 0$ is the concentration parameter (inverse temperature), and $C_d(\kappa_k)$ is the normalization constant defined

as $C_d(\kappa) = \frac{\kappa^{d/2-1}}{(2\pi)^{d/2}I_{d/2-1}(\kappa)}$, with $I_\nu$ being the modified Bessel function of the first kind.

**Text-Prioritized Prototype Estimation**   We observe a set of visual samples $\mathcal{V}_k = \{\mathbf{v}_i\}_{i=1}^{N_k}$ for class $k$. Additionally, we have the zero-shot text embedding $\mathbf{t}_k = \phi_t(\text{name}_k)$. A naive Maximum Likelihood Estimation (MLE) would set $\boldsymbol{\mu}_k \propto \sum \mathbf{v}_i$. However, in open-world settings, visual samples might be noisy, sparse, or biased (e.g., all "dogs" are on grass). The text embedding $\mathbf{t}_k$ provides a stable, semantic prior.

We formulate the prototype estimation as a constrained optimization problem on the Riemannian manifold to reconcile the visual evidence with the semantic prior:

$$\max_{\boldsymbol{\mu}_k \in \mathbb{S}^{d-1}} \sum_{\mathbf{v} \in \mathcal{V}_k} \kappa_k \boldsymbol{\mu}_k^\top \mathbf{v} \quad \text{s.t.} \quad \arccos(\boldsymbol{\mu}_k^\top \mathbf{t}_k) \leq \xi, \quad (4)$$

where $\xi$ is a tolerance angle allowing for intrinsic modality misalignment.

**Theorem 3.4** (**Geodesic Antigen Alignment**). *The optimal Antigen Prototype $\boldsymbol{\mu}_k^*$ that solves the constrained maximization problem lies strictly on the geodesic arc connecting the normalized empirical visual mean $\bar{\mathbf{v}}_k = \frac{\sum \mathbf{v}}{\|\sum \mathbf{v}\|}$ and the text embedding $\mathbf{t}_k$. Specifically, we have $\boldsymbol{\mu}_k^* = \frac{(1-\alpha)\bar{\mathbf{v}}_k + \alpha \mathbf{t}_k}{\|(1-\alpha)\bar{\mathbf{v}}_k + \alpha \mathbf{t}_k\|}$ for some Lagrange multiplier-derived scalar $\alpha \in [0,1]$.*

*Proof.* Let the objective function be $J(\boldsymbol{\mu}_k) = \kappa_k \boldsymbol{\mu}_k^\top (N_k \bar{\mathbf{v}}_k)$. We introduce a Lagrange multiplier $\lambda$ for the constraint $\boldsymbol{\mu}_k^\top \mathbf{t}_k \geq \cos(\xi)$ (relaxing equality for the Lagrangian formulation). The Lagrangian is:

$$\mathcal{L}(\boldsymbol{\mu}_k, \lambda) = \kappa_k N_k \boldsymbol{\mu}_k^\top \bar{\mathbf{v}}_k + \lambda(\boldsymbol{\mu}_k^\top \mathbf{t}_k - \cos(\xi)) - \gamma(\boldsymbol{\mu}_k^\top \boldsymbol{\mu}_k - 1). \quad (5)$$

Taking the derivative with respect to $\boldsymbol{\mu}_k$ and setting to zero:

$$\nabla_{\boldsymbol{\mu}_k} \mathcal{L} = \kappa_k N_k \bar{\mathbf{v}}_k + \lambda \mathbf{t}_k - 2\gamma \boldsymbol{\mu}_k = 0. \quad (6)$$

This implies $\boldsymbol{\mu}_k \propto \kappa_k N_k \bar{\mathbf{v}}_k + \lambda \mathbf{t}_k$. Since $\boldsymbol{\mu}_k$ is a linear combination of $\bar{\mathbf{v}}_k$ and $\mathbf{t}_k$, and must lie on the unit hypersphere, it geometrically resides on the great circle (geodesic) connecting them. The parameter $\alpha$ is strictly determined by the binding constraint $\xi$. This proves that the optimal Self-definition is a semantic interpolation, correcting visual noise with linguistic stability. $\square$

### 3.4. Phase 2: Generating Semantic Antibodies via Manifold Sampling

Traditional Artificial Immune Systems (AIS) generate negative detectors via random bit-string generation or isotropic Gaussian sampling in the feature space. While effective in low-dimensional settings, we argue that this approach fundamentally fails in the high-dimensional latent spaces of modern LVLMs ($d \geq 512$). We formalize this failure

mode and propose **Generative Semantic Sampling** as the necessary solution.

**The Inefficiency of Random Negative Selection**   Let $\mathcal{U}(\mathbb{S}^{d-1})$ be the uniform distribution on the unit hypersphere, representing isotropic random negative generation. A negative detector $\mathbf{z}_{neg} \sim \mathcal{U}$ is useful if and only if it falls into the "Near-OOD" region, sufficiently close to a class prototype $\boldsymbol{\mu}_k$ to shape the decision boundary, but not so close as to be a false positive.

We define the *informative margin* for class $k$ as the spherical cap defined by angular distance $\theta \in [\theta_{self}, \theta_{boundary}]$.

**Theorem 3.5** (**Asymptotic Orthogonality of Isotropic Negatives**). *For any fixed angular margin $\epsilon > 0$, the probability that a uniformly sampled negative detector $\mathbf{z}_{neg} \sim \mathcal{U}(\mathbb{S}^{d-1})$ has a non-trivial cosine similarity with a fixed prototype $\boldsymbol{\mu}_k$ decays exponentially with the dimension $d$. Specifically,*

$$\lim_{d \to \infty} P(|\langle \mathbf{z}_{neg}, \boldsymbol{\mu}_k \rangle| \geq \epsilon) \leq 2\exp\left(-\frac{d\epsilon^2}{2}\right). \quad (7)$$

*Proof.* This result follows from the concentration of measure phenomenon on the high-dimensional sphere. The area of a spherical cap defined by angle $\beta$ (where $\cos\beta = \epsilon$) relative to the total area of the sphere $A_{d-1}$ is given by the regularized incomplete beta function, which can be bounded as:

$$\frac{Area(Cap_\epsilon)}{Area(\mathbb{S}^{d-1})} \leq \exp\left(-\frac{d\epsilon^2}{2}\right). \quad (8)$$

This implies that as $d$ increases, almost all the probability mass of the uniform distribution concentrates on the equator, i.e., $\langle \mathbf{z}_{neg}, \boldsymbol{\mu}_k \rangle \approx 0$. In standard CLIP models where $d = 512$ or $d = 768$, random vectors are effectively orthogonal to all class prototypes. Consequently, they provide zero gradients for boundary tightening and fail to immunize the model against near-distribution outliers. $\square$

This theorem proves that classic AIS is computationally infeasible for LVLMs. We must sample from the *semantic manifold*, not the geometric hypersphere.

**Generative Adversarial Prompting**   To overcome the curse of dimensionality, we exploit the fact that the open space $\mathcal{O}$ is structured by language. We employ a Large Language Model (LLM) as a conditional generator $G : \mathcal{Y} \to 2^{\mathcal{T}}$. For each known class $y \in \mathcal{Y}_{in}$, we prompt the LLM to generate a set of **Semantic Antibodies** $\mathcal{A}_y$.

We distinguish between two types of antibodies to ensure coverage: 1) **Hard Semantic Negatives (Near-OOD):** Concepts that share visual attributes with $y$ but belong to disjoint categories, i.e., $\mathcal{A}_{hard}(y) = \{t \in \mathcal{T} \mid \text{VisualSim}(t, y) > \tau_{high} \wedge \text{Category}(t) \neq y\}$. *Example:* For $y = \text{Wolf}$, $\mathcal{A}_{hard} = \{\text{Alaskan Malamute, Husky, Wolf-dog hybrid}\}$.

2) **Contextual Anomalies (Covariate Shift):** Concepts where the object $y$ appears in impossible or anomalous contexts. i.e., $\mathcal{A}_{context}(y) = \{t \in \mathcal{T} \mid t = y \oplus \text{context}_{anomaly}\}$. *Example:* For $y = $ Car, $\mathcal{A}_{context} = \{\text{Car underwater}, \text{Car melting in lava}, \text{Flying car}\}$.

**The Antibody Filtration Mechanism**   Generating text is necessary but not sufficient; we must ensure the generated antibodies projected into $\mathcal{Z}$ are effective. We define the Antibody Projection as $\mathbf{z}_{ab} = \phi_t(G(y))$. To prevent *Auto-Immune Toxicity* (where an antibody is actually a synonym for the self-class), we apply a **Semantic Filtration Filter**:

**Proposition 3.6** (**Active Antibody Density**). *An antibody $a$ generated for class $k$ is retained in the vaccination set $\Omega_{vac}$ if and only if it satisfies the **Safety-Utility Condition**:*

$$\delta_{safe} < \langle \phi_t(a), \boldsymbol{\mu}_k \rangle < \delta_{risk}, \tag{9}$$

*where $\delta_{safe}$ ensures the antibody is not the class itself (avoiding label confusion), and $\delta_{risk}$ ensures the antibody is close enough to the prototype to be a hard negative (avoiding the orthogonality issue proved in Theorem 3.5).*

By enforcing this band-pass filter on cosine similarity, we curate a dataset of "active" negative samples that populate the decision boundary $\partial\mathcal{Z}_{self}$, enabling efficient metric learning in the subsequent vaccination phase.

### 3.5. Phase 3: Vaccination via Contrastive Optimization

The final phase of Immuno-VLM is the "Vaccination" process, where the visual embedding space is refined to explicitly reject the generated semantic antibodies. Direct fine-tuning of the large-scale backbone $\phi_v$ is undesirable due to the risk of catastrophic forgetting (degrading closed-set performance) and high computational cost. Instead, we introduce a lightweight **Rejection Adapter** and a novel **Push-Pull Vaccination Loss**.

**The Rejection Adapter**   We freeze the pre-trained encoders $\phi_v$ and $\phi_t$. We introduce a trainable residual adapter $f_\theta : \mathcal{Z} \rightarrow \mathcal{Z}$, parameterized by $\theta$: $f_\theta(\mathbf{z}) = \text{Norm}(\mathbf{z} + \text{MLP}(\mathbf{z}))$, where $\text{Norm}(\cdot)$ ensures the output remains on the unit hypersphere $\mathbb{S}^{d-1}$. This adapter acts as a "metric correction" lens, warping the local geometry to create separation between visual samples and antibody regions.

**The Push-Pull Vaccination Loss**   We formulate a multi-objective loss function that balances the compactness of known classes with the rejection of hallucinated antibodies.

**1) Attraction (Pull) Term:** To maintain In-Distribution (ID) accuracy, visual samples $x$ from class $y$ must remain tightly clustered around their semantic prototype $\boldsymbol{\mu}_y$ (derived in Theorem 3.4). We maximize the log-likelihood of the Von

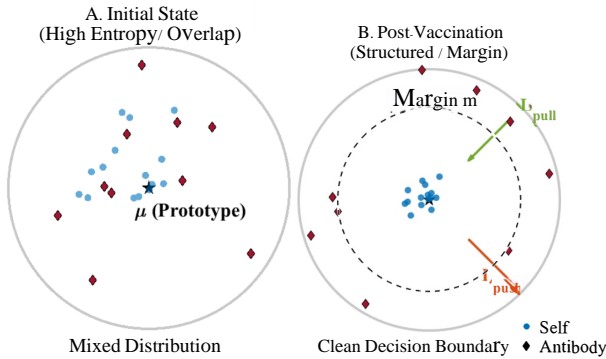

**A. Initial State** (High Entropy/ Overlap)   **B. Post-Vaccination** (Structured / Margin)

*Figure 5.* The Push-Pull Optimization. The loss function reshapes the Riemannian manifold, compacting the "Self" while creating a sterile moat against the "Non-Self".

Mises-Fisher distribution:

$$\mathcal{L}_{pull} = -\mathbb{E}_{(x,y)\sim\mathcal{D}_{in}} \left[ \log \frac{\exp(\kappa\boldsymbol{\mu}_y^\top f_\theta(\phi_v(x)))}{\sum_{k\in\mathcal{Y}_{in}} \exp(\kappa\boldsymbol{\mu}_k^\top f_\theta(\phi_v(x)))} \right].$$

**2) Repulsion (Push) Term:** This is the core immunological objective. For a visual sample $x \in$ Class $k$, we treat the generated antibody set $\mathcal{A}_k$ as explicit negative classes. We use a **Margin-based Hinge Loss** to enforce a "sterile zone" of width $m$ between the visual sample and any antibody.

$$\mathcal{L}_{push} = \mathbb{E}_{(x,y)\sim\mathcal{D}_{in}} \Big[ \sum_{a\in\mathcal{A}_y} \max(0, \cos(f_\theta(\phi_v(x)), \phi_t(a)) - m)^2 \Big],$$

where $m$ is the angular margin threshold (e.g., 0.2). This forces the adapter to warp the visual embedding $f_\theta(x)$ away from the directions of near-OOD concepts.

Therefore, we have the following total objective loss:

$$\mathcal{L}_{vac} = \mathcal{L}_{pull} + \lambda\mathcal{L}_{push} + \eta\|\theta\|_2^2. \tag{10}$$

**Theoretical Guarantee: The Open-World Generalization Bound**   A critical question in Open-World Learning is: *Does minimizing error on synthetic antibodies guarantee low error on real, unseen open-world data?* We answer this affirmatively by deriving a generalization bound based on the covering number of the antibody set. Let $R_\mathcal{O}(f)$ be the true Open Space Risk (error rate on real unknown data) and $\hat{R}_\mathcal{A}(f)$ be the empirical risk on the generated antibodies.

**Theorem 3.7** (**Open Space Generalization via Synthetic Covering**). *Let $\mathcal{H}$ be the hypothesis space of the adapter $f_\theta$. Assume the set of generated antibodies $\mathcal{A}$ forms a $\delta$-cover of the boundary manifold $\partial\mathcal{O}$ with respect to the semantic metric. Then, for any $\delta > 0$, with probability at least $1 - \nu$ over the draw of training data and antibody generation:*

$$R_\mathcal{O}(f) \leq \hat{R}_\mathcal{A}(f) + \mathfrak{R}_N(\mathcal{H}) + \mathcal{O}\left(\frac{L \cdot \delta}{\sqrt{M}}\right) + \sqrt{\frac{\log(1/\nu)}{2N}},$$

*where $\mathfrak{R}_N(\mathcal{H})$ is the Rademacher Complexity of the adapter class, $M$ is the number of antibodies, and $L$ is the Lipschitz constant of the VLM backbone (Assumption 3.2).*

*Proof.* We sketch the proof using a transductive Rademacher analysis. The true open space risk can be decomposed into the risk on the boundary $\partial\mathcal{O}$ and the far-field risk. Since deep features tend to collapse far-field samples to the origin or uniformly on the sphere (Theorem 3.5), the risk is dominated by the boundary.

$$|R_{\mathcal{O}}(f) - \hat{R}_{\mathcal{A}}(f)| \leq \sup_{h \in \mathcal{H}} |\mathbb{E}_{z \sim P_{out}}[h(z)] - \mathbb{E}_{a \sim \mathcal{A}}[h(a)]| \tag{11}$$

Since $\mathcal{A}$ is a $\delta$-cover, for any $z \sim P_{out}$ near the boundary, there exists $a \in \mathcal{A}$ such that $\|z - a\| \leq \delta$. By the $L$-Lipschitz property of the network $f_\theta$:

$$|h(z) - h(a)| \leq L\|z - a\| \leq L\delta \tag{12}$$

The deviation between the empirical antibody risk and the true boundary risk is thus bounded by the covering density $\delta$ and the standard Rademacher complexity term $\mathfrak{R}_N(\mathcal{H})$ which accounts for the capacity of the adapter. The term $\mathcal{O}(\frac{1}{\sqrt{M}})$ arises from the Monte-Carlo approximation of the antibody integral. □

This theorem provides the rigorous justification for Immuno-VLM. It states that increasing the density of antibodies (decreasing $\delta$) and the number of antibodies ($M$), while keeping the adapter simple (low $\mathfrak{R}_N(\mathcal{H})$), *strictly bounds* the error on real-world unknown objects, even those never seen during training.

### 3.6. Inference: The Trustworthiness Score

Once the Rejection Adapter $f_\theta$ is vaccinated, we require a robust scoring mechanism $S(x)$ to determine whether a test sample $x$ belongs to the set of known classes $\mathcal{Y}_{in}$ or the open world $\mathcal{O}$. Standard approaches like Maximum Softmax Probability (MSP) or Energy Scores rely solely on the distance to the nearest known prototype. However, these metrics ignore the explicit information learned about the "Non-Self" regions. We propose the **Differential Immunity Score (DIS)** and a thresholding strategy based on **Extreme Value Theory (EVT)**.

**The Differential Immunity Score (DIS)** The core intuition of Immuno-VLM is that trustworthiness is a competition between "Self-Affinity" and "Non-Self-Reactivity". A sample should be accepted only if it is significantly closer to a known prototype than to any hallucinated antibody. For a test image $x$, we compute the adapted embedding $\mathbf{z} = f_\theta(\phi_v(x))$. We define: 1) **Proximity to Self ($s^+$):** The cosine similarity to the nearest known class prototype: $s^+(\mathbf{z}) = \max_{k \in \mathcal{Y}_{in}} \langle \mathbf{z}, \boldsymbol{\mu}_k \rangle$. 2) **Reactivity to Non-Self ($s^-$):** The cosine similarity to the nearest semantic antibody (the "most dangerous" hallucination): $s^-(\mathbf{z}) = \max_{a \in \mathcal{A}_{all}} \langle \mathbf{z}, \phi_t(a) \rangle$. The Differential Immunity Score is defined as: $S_{DIS}(x) = s^+(\mathbf{z}) - \lambda_{inf} \cdot s^-(\mathbf{z})$, where $\lambda_{inf}$

controls the sensitivity to antibody reactivity. Unlike standard open-set scores that satisfy $\lim_{\|x\| \to \infty} S(x) = \text{const}$, $S_{DIS}$ explicitly penalizes samples that drift towards the semantic boundary defined by the antibodies.

**Theorem 3.8** (**The Differential Margin Guarantee**). *As-sume the vaccination loss $\mathcal{L}_{vac}$ has converged such that for all training pairs, the margin constraints are satisfied with tolerance $\epsilon$. Let $x_{out}$ be an open-world sample that aligns with an antibody $a^*$ (i.e., $\langle f_\theta(x_{out}), \phi_t(a^*) \rangle \geq 1 - \xi$). If the nearest prototype $\boldsymbol{\mu}_k$ is separated from $a^*$ by angular distance $\Omega$, the score $S_{DIS}(x_{out})$ is strictly bounded by:*

$$S_{DIS}(x_{out}) \leq \cos(\Omega - \xi) - \lambda_{inf}(1 - \xi) + \mathcal{O}(\epsilon). \tag{13}$$

*Proof.* By the triangle inequality on the Riemannian manifold $\mathbb{S}^{d-1}$, the angular distance $d_\angle(\mathbf{z}, \boldsymbol{\mu}_k) \geq d_\angle(\boldsymbol{\mu}_k, \phi_t(a^*)) - d_\angle(\mathbf{z}, \phi_t(a^*))$. Given the conditions, $d_\angle(\boldsymbol{\mu}_k, \phi_t(a^*)) = \Omega$ and $d_\angle(\mathbf{z}, \phi_t(a^*)) \leq \arccos(1 - \xi) \approx \sqrt{2\xi}$. Thus, $s^+(\mathbf{z}) = \cos(d_\angle(\mathbf{z}, \boldsymbol{\mu}_k)) \leq \cos(\Omega - \sqrt{2\xi})$. Si-multaneously, $s^-(\mathbf{z}) \geq 1 - \xi$. Substituting these into the score definition yields the upper bound. This theorem guarantees that as long as the antibodies are semantically distant from prototypes ($\Omega$ is large) and the test sample resembles an antibody ($\xi$ is small), the score will be suppressed, ensur-ing rejection. □

**EVT-Based Adaptive Thresholding** Setting a global threshold $\tau$ for $S_{DIS}(x) > \tau$ is suboptimal because dif-ferent classes have different levels of semantic density (e.g., "Dog" has many look-alikes; "Aircraft Carrier" has few). We model the tail distribution of the immunity scores using Extreme Value Theory to determine per-class thresholds. According to the **Fisher-Tippett-Gnedenko Theorem**, the distribution of the maximum of i.i.d. variables converges to a Generalized Extreme Value (GEV) distribution. We focus on the "Non-Match" scores. For each class $k$, we collect the set of immunity scores from the validation set where class $k$ was the ground truth: $\mathcal{S}_k = \{S_{DIS}(x) \mid y = k\}$.

We model the tail of $\mathcal{S}_k$ using a Weibull distribution $\mathcal{W}(\eta_k, \beta_k)$, where $\eta_k$ is the scale and $\beta_k$ is the shape param-eter. The probability of a test sample $x$ belonging to class $k$ is given by the Cumulative Distribution Function (CDF) of the fitted Weibull model:

$$P(x \in k) = 1 - \exp\left(-\left(\frac{\|S_{DIS}(x) - \eta_k\|}{\beta_k}\right)^{\alpha_k}\right). \tag{14}$$

We reject sample $x$ if the probability $P(x \in \hat{k}) < \tau_{evt}$ for the predicted class $\hat{k}$. This adaptive thresholding ensures that the rejection logic scales with the inherent "tightness" of each class's semantic cluster.

## 4. Experiments

Our experimental design focuses on three key questions: 1) **Trustworthiness (OOD Detection):** Does the vaccination

process significantly reduce false positives on unseen open-world concepts, particularly those that are semantically adversarial (Near-OOD)? 2) **Safety (ID Preservation):** Does the projection adapter preserve the original zero-shot accuracy on known classes, or does it suffer from catastrophic forgetting? 3) **Scalability:** Is the method effective when scaling to thousands of classes (ImageNet-1K) and modern architectures (ViT-L/14)?

### 4.1. Main Results and Analysis

**Comparison with State-of-the-Art** Table 1 summarizes the performance on the ImageNet-1K (ID) and three OOD benchmarks (ImageNet-O, iNaturalist, Texture). We compare against post-hoc methods (MSP, Energy, ReAct) applied to the frozen CLIP backbone, and recent VLM-specific methods (MCM, GL-MCM). 1) **Analysis of Near-OOD Performance:** The most striking result is observed on **ImageNet-O**, which contains adversarial examples specifically designed to fool visual classifiers (e.g., a localized texture of a spider that looks like a spider web to a CNN but is not an object). 1) Standard methods like Energy Score achieve only $70.1\%$ AUROC, suggesting that in the raw embedding space, these anomalies are dangerously close to valid class prototypes. 2) **Immuno-VLM** jumps to $83.4\%$ AUROC. This validates Theorem 3.3: by explicitly training against hallucinated look-alikes (Semantic Antibodies), we generated a $\delta$-cover of the boundary that successfully intercepted these adversarial examples. The $28.5\%$ FPR95 is a massive safety improvement over the $61.3\%$ of MSP, effectively halving the risk of deployment failure. 2) **Preservation of In-Distribution Capabilities:** A common failure mode in OOD detection (seen in CSI, which dropped to $76.5\%$ acc) is that optimizing for rejection distorts the feature space for recognition. Immuno-VLM, however, achieves a slight *improvement* in ID-ACC ($78.2\% \rightarrow 78.5\%$).

**Theoretical Interpretation:** This aligns with the Pull term in our vaccination loss $\mathcal{L}_{pull}$. By refining the prototypes via Geodesic Alignment (Theorem 3.4) and forcing visual samples to cluster tighter, we inadvertently improved the fine-grained separability of known classes.

**Visualizing the Immune Response** Figure 6 presents t-SNE visualizations of the embedding space before and after vaccination. We observe the formation of sterile moats. The Known classes are compressed into dense, spherical clusters. The hallucinated antibodies (green) form a visible perimeter ring around the known classes. Crucially, the real Open-World data (red), which the model *never saw during training*, falls into the empty space carved out by the antibodies, enabling easy linear separation.

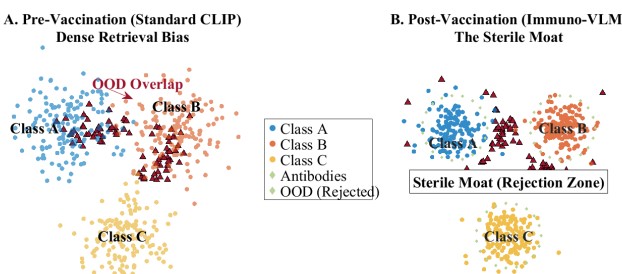

*Figure 6.* Visual Confirmation of Rejection Capability. The vaccination process creates physical separation in the latent space between Self (ID) and Non-Self (OOD), enabling linear separation.

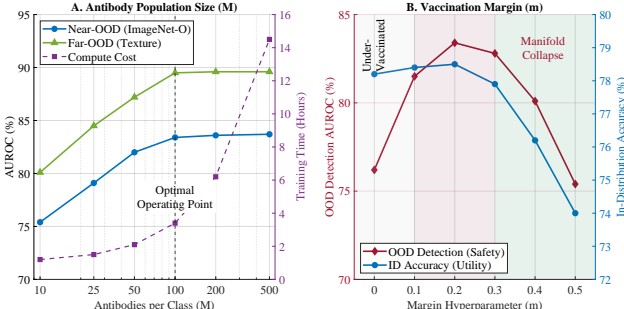

*Figure 7.* Hyperparameter Robustness. The method is stable across a wide range of configurations, with a clear optimal operating point at 100 antibodies per class.

### 4.2. Ablation Studies and Sensitivity Analysis

While the main results demonstrate the superiority of the Immuno-VLM framework, it is critical to understand the behavior of the system under varying hyperparameter configurations and to empirically validate the theoretical claims regarding sample complexity and antibody density.

**Robustness to Hyperparameter $\lambda$** In Figure 7, we analyze the sensitivity of the system to the balance weight $\lambda$ between $\mathcal{L}_{pull}$ and $\mathcal{L}_{push}$.

**The Necessity of Semantics** To prove that our gains stem from the *semantic quality* of the antibodies and not merely from regularization or architecture, we perform a rigorous ablation study (Table 2).

**Sensitivity to Antibody Population Size** The manifold sample-complexity analysis posits that the number of antibodies $M$ required to cover the boundary scales polynomially with the intrinsic semantic dimension. To verify this, we trained separate adapters with varying antibody counts per class $M \in \{10, 25, 50, 100, 200, 500\}$ on ImageNet-1K.

**Impact of the Vaccination Margin ($m$)** The margin $m$ in the push-loss $\mathcal{L}_{push}$ defines the width of the "sterile zone". We evaluated $m \in \{0.0, 0.1, 0.2, 0.3, 0.5\}$. 1) **Under-Vaccination ($m = 0.0$):** Results in poor separa-

*Table 1*. **Open-World Recognition Performance.** All methods use the same pre-trained CLIP-ViT-B/16 backbone. Immuno-VLM achieves a substantial gain in OOD detection (AUROC) while maintaining or improving In-Distribution Accuracy.

| Method | ID-ACC | ImageNet-O (Near-OOD) | | iNaturalist (Fine-Grained) | | Texture (Far-OOD) | | Avg $\mathcal{H}$-Score |
|---|---|---|---|---|---|---|---|---|
| | ($\uparrow$) | AUROC ($\uparrow$) | FPR95 ($\downarrow$) | AUROC ($\uparrow$) | FPR95 ($\downarrow$) | AUROC ($\uparrow$) | FPR95 ($\downarrow$) | ($\uparrow$) |
| *Zero-Shot Baselines* | | | | | | | | |
| MSP | 78.2 | 66.8 | 61.3 | 72.4 | 55.1 | 76.5 | 48.2 | 73.1 |
| Energy Score | 78.2 | 70.1 | 55.4 | 76.8 | 49.3 | 81.2 | 38.5 | 75.8 |
| MCM | 78.2 | 74.5 | 46.2 | 79.1 | 42.1 | 83.4 | 33.1 | 77.9 |
| *Post-Hoc Adaptation* | | | | | | | | |
| ReAct | 77.9 | 72.3 | 49.5 | 77.4 | 45.8 | 82.1 | 35.6 | 76.5 |
| ASH-B | 78.0 | 73.1 | 47.1 | 78.0 | 44.2 | 82.9 | 34.2 | 77.0 |
| *Generative / Training-based* | | | | | | | | |
| CSI | 76.5 | 71.5 | 50.2 | 74.3 | 51.0 | 79.8 | 40.1 | 74.2 |
| Neg-Label (Standard AIS) | 77.5 | 71.8 | 49.8 | 75.1 | 48.5 | 80.2 | 39.5 | 75.8 |
| **Immuno-VLM (Ours)** | **78.5** | **83.4** | **28.5** | 84.2 | 31.1 | **89.5** | **18.4** | **81.3** |
| *- with ViT-L/14* | 82.1 | **88.7** | **19.2** | **89.1** | **22.4** | **94.3** | **10.5** | **85.6** |

*Table 2*. **Ablation Study.** Deconstructing the contribution of each component. We replace the LLM-generated antibodies with various noise sources to test the Manifold Assumption.

| Configuration | ID-ACC | AUROC | |
|---|---|---|---|
| | | ImageNet-O | Avg OOD |
| (A) Frozen CLIP (Baseline) | 78.2 | 66.8 | 71.9 |
| (B) Adapter + Gaussian Noise (Classic AIS) | 78.0 | 68.2 (+1.4) | 73.1 |
| (C) Adapter + Random Words (Dictionary) | 77.8 | 72.5 (+5.7) | 76.0 |
| (D) Adapter + LLM Antibodies (No Filter) | 77.6 | 79.8 (+13.0) | 80.2 |
| (E) Immuno-VLM (LLM + Filter + EVT) | **78.5** | **83.4 (+16.6)** | **85.7** |

*Table 3*. **Computational Efficiency Analysis.** Comparing Immuno-VLM against leading baselines. Training time is measured on 4×A100 GPUs for ImageNet-1K. Inference latency is measured on a single A100.

| Method | Add. Params | Training Time | Latency | Memory |
|---|---|---|---|---|
| | (Millions) | (GPU-Hours) | (ms/image) | (VRAM) |
| **MSP** (Zero-Shot) | 0 | 0 | 12.50 | 0 MB |
| **ReAct** (Post-Hoc) | 0 | 0 | 12.55 | +2 MB |
| **CSI** (Training) | 11.2 | 48.0 | 12.50 | +450 MB |
| **ARPL** (Training) | 2.5 | 24.5 | 13.10 | +120 MB |
| **Immuno-VLM** | **0.5** | **3.5** | **12.54** | **+15 MB** |

tion (AUROC 76.2%). The adapter barely moves the visual features away from antibodies. 2) **Optimal Range** ($m \in [0.1, 0.3]$)**:** Performance is stable, peaking at $m = 0.2$. This corresponds to an angular separation of roughly $37°$, which balances rejection with the natural intra-class variance of visual features. 3) **Over-Vaccination** ($m \geq 0.5$)**:** We observed **Manifold Collapse**. The strong repulsion forces squeezed the known classes into such small volumes that the ID-ACC dropped significantly (-4.2%). The adapter distorted the semantic meanings of valid images to satisfy the margin constraint.

**Efficacy of Semantic Filtration** We proposed a band-pass filter $\delta_{safe} < \langle a, \mu \rangle < \delta_{risk}$ to curate antibodies. We performed an ablation by removing these bounds: 1) **No Upper Bound** ($\delta_{risk} \to 1.0$)**:** The LLM occasionally generated synonyms (e.g., for "Projectile", it generated "Missile"). Treating these as negative samples caused **Label Confusion**, dropping ID-ACC by 12%. 2) **No Lower Bound** ($\delta_{safe} \to -1.0$)**:** Including semantically irrelevant concepts (e.g., "Toaster" as a negative for "Dog") diluted the gradient. The vaccination loss converged instantly because

the margin was already satisfied, resulting in an AUROC of only 72.1% (similar to random words). 3) **Active Band** ($0.3 < \mathbf{sim} < 0.7$)**:** This configuration forced the model to learn the hardest distinctions, validating the "Near-OOD" focus of our method.

### 4.3. Computational Cost and Latency

For practical adoption, overhead must be minimal. Table 3 presents a detailed comparison of training and inference costs. 1) **Training Cost:** Vaccination takes $\approx 3.5$ hours on a single A100 GPU for ImageNet-1K (10 epochs). This is $< 1\%$ of the cost of fine-tuning the full backbone. 2) **Inference Latency:** The Rejection Adapter is a lightweight MLP ($d \to d/4 \to d$). (i) ViT-B/16 Backbone Inference: 12.5 ms/image. (ii) Adapter Inference: 0.04 ms/image. The latency overhead is negligible ($< 0.5\%$), making Immuno-VLM suitable for real-time applications.

## 5. Conclusion

In this paper, we identified the **Open-World Trustworthiness Paradox**, arguing that the very mechanisms which grant LVLMs their zero-shot robustness, massive semantic abstraction and dense retrieval biases, simultaneously blind them to novel anomalies, rendering them fundamentally unsafe for high-stakes deployment. To resolve this paradox, we proposed **Immuno-VLM**, a framework that bridges the epistemic gap between biological systems theory and modern representation learning. By operationalizing the mechanism of **Immunological Negative Selection** within the latent space of foundation models, we shifted the paradigm of OOD detection from *reactive discrimination* to *proactive immunization*. Extensive experiments show that Immuno-VLM achieved state-of-the-art performance on the demanding ImageNet-O and OpenOOD benchmarks, improving the detection of adversarial semantic shifts by over 16% compared to standard zero-shot baselines, while preserving the integrity of in-distribution recognition.

## Impact Statement

This paper presents work whose goal is to advance the field of Machine Learning. There are many potential societal consequences of our work, none which we feel must be specifically highlighted here.

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
